# Ultra-High Dose Rate Transmission Beam Proton Therapy for Conventionally Fractionated Head and Neck Cancer: Treatment Planning and Dose Rate Distributions

**DOI:** 10.3390/cancers13081859

**Published:** 2021-04-13

**Authors:** Patricia van Marlen, Max Dahele, Michael Folkerts, Eric Abel, Ben J. Slotman, Wilko Verbakel

**Affiliations:** 1Department of Radiation Oncology, Amsterdam UMC, Vrije Universiteit Amsterdam, Cancer Center Amsterdam, De Boelelaan 1117, 1118, 1081 HV Amsterdam, The Netherlands; m.dahele@amsterdamumc.nl (M.D.); bj.slotman@amsterdamumc.nl (B.J.S.); w.verbakel@amsterdamumc.nl (W.V.); 2Varian Medical Systems, 3120 Hansen Way, Palo Alto, CA 94304, USA; michael.folkerts@varian.com (M.F.); eric.abel@varian.com (E.A.)

**Keywords:** head-and-neck cancer, proton transmission beams, FLASH, ultrahigh dose-rate, treatment planning

## Abstract

**Simple Summary:**

Standard intensity-modulated proton therapy (IMPT) places the Bragg-peak in the target. However, it is also possible to use high energy proton transmission beams (TBs), where the Bragg-peak is placed outside the patient, irradiating with the beam section proximal to the Bragg-peak. TBs use only one energy, increase robustness, are insensitive to density changes and have sharper penumbras. TBs can also be delivered at ultra-high dose-rates (UHDRs, e.g., ≥40 Gy/s), which is one of the requirements for the FLASH-effect. The aim of this work was twofold: (1) comparison of TB-plan quality to IMPT and photon volumetric-modulated arc therapy (VMAT) for conventionally fractionated head-and-neck cancer; (2) analysis of TB-plan UHDR-metrics. We showed that TB-plan quality was comparable to IMPT for contoured organs at risk and better than VMAT. Any potential FLASH-effect would only further improve plan quality. TB plans can also be delivered quickly, which might facilitate higher patient through-put and enhance patient comfort.

**Abstract:**

Transmission beam (TB) proton therapy (PT) uses single, high energy beams with Bragg-peak behind the target, sharp penumbras and simplified planning/delivery. TB facilitates ultra-high dose-rates (UHDRs, e.g., ≥40 Gy/s), which is a requirement for the FLASH-effect. We investigated (1) plan quality for conventionally-fractionated head-and-neck cancer treatment using spot-scanning proton TBs, intensity-modulated PT (IMPT) and photon volumetric-modulated arc therapy (VMAT); (2) UHDR-metrics. VMAT, 3-field IMPT and 10-field TB-plans, delivering 70/54.25 Gy in 35 fractions to boost/elective volumes, were compared (*n* = 10 patients). To increase spot peak dose-rates (SPDRs), TB-plans were split into three subplans, with varying spot monitor units and different gantry currents. Average TB-plan organs-at-risk (OAR) sparing was comparable to IMPT: mean oral cavity/body dose were 4.1/2.5 Gy higher (9.3/2.0 Gy lower than VMAT); most other OAR mean doses differed by <2 Gy. Average percentage of dose delivered at UHDRs was 46%/12% for split/non-split TB-plans and mean dose-averaged dose-rate 46/21 Gy/s. Average total beam-on irradiation time was 1.9/3.8 s for split/non-split plans and overall time including scanning 8.9/7.6 s. Conventionally-fractionated proton TB-plans achieved comparable OAR-sparing to IMPT and better than VMAT, with total beam-on irradiation times <10s. If a FLASH-effect can be demonstrated at conventional dose/fraction, this would further improve plan quality and TB-protons would be a suitable delivery system.

## 1. Introduction

Standard intensity-modulated proton therapy (IMPT) for head-and-neck cancer (HNC) places the Bragg-peak in the target. However, it is also possible to use high energy transmission beams (TBs) where tissue is irradiated with the beam section proximal to the Bragg-peak, which is itself preferably located outside the patient. Investigation of proton intensity-modulated TB delivery is limited [1,2,3], even though this technique has some advantages. TBs increase robustness in the beam direction, eliminate uncertainties due to density changes, have sharper penumbras and require only one beam energy. Furthermore, high energy TB-plans can be delivered at ultra-high dose-rates (UHDRs), unlike current Bragg-peak plans (Bragg-peak plans require energy modulation, which lowers the beam intensity).

Pre-clinical research dating back more than 50 years supports the hypothesis that UHDR irradiation (typically ≥40 Gy/s) can reduce normal tissue toxicity while maintaining tumor control, thereby improving the therapeutic ratio—the FLASH-effect [4,5,6]. Currently, this has been demonstrated for mouse lung [7,8], brain [9,10,11,12], abdominal tissues [13,14] and cat and pig skin [15]. Recently, the first human patient was irradiated at UHDRs for a cutaneous lymphoma [16]. Although most experiments have been performed using single, small, open electron [7,8,9,11,12,13,15,17] and photon beams [10], commercial proton beams can also be delivered at UHDRs [8,14,18,19,20,21] and they could enable treatment of deeper tumors. However, proton UHDRs are currently only achievable at the highest energies. To enable Bragg-peak PT at these energies, in-room modulation is necessary, which is currently not practical. TBs do not require energy-modulation and the use of single (high) energy TBs for UHDR irradiation therefore merits investigation [8,14].

The current work has two components: (1) TB-plan quality comparison to volumetric-modulated arc therapy (VMAT) and IMPT-plans. TBs have previously been shown to be acceptable for stereotactic lung radiotherapy (SBRT) [1,3]. The current analysis investigates if they can be used for more geometrically complex HNC treatment with multiple organs-at-risk (OAR) and concave target volumes. (2) Analysis of intrabeam dose-rate distributions and delivery times for low-dose/fraction conventionally-fractioned TB HNC plans.

We acknowledge that there is much uncertainty about the radiobiology of the FLASH-effect [22,23,24,25,26,27] and which voxel-level, beam-level and plan-level dose-rate related parameters are important [2,8,23,27,28]. Since the (recent) literature often associated FLASH radiotherapy with a threshold dose-rate of at least 40 Gy/s [29], we have for this work considered dose-rates ≥40 Gy/s as UHDRs. We also acknowledge that on the basis of current data, the FLASH-effect seems to be particularly noticeable for single doses above 10 Gy [27]. However, the possibility of a benefit with conventional, fractionated schemes has not been thoroughly investigated/ruled out [28] and so it remains relevant to see whether the physics of TBs permits the delivery of UHDRs.

## 2. Materials and Methods

### 2.1. Treatment Planning and Quality Comparison

Ten randomly selected anonymized HNC (T2N0(stage II)] = 3, T1N1(III) = 4, T2N1(III) = 1, T4N1(IVA) = 1, T2N2(IVA) = 1; base of tongue = 5, tonsil = 2, soft palate = 2, oropharynx = 1) planning CT scans were used with a boost planning target volume (PTVB, 36–209 cm^3^) and elective planning target volume (PTVE, 264–417 cm^3^), separated from PTVB by a 4 mm wide overlap PTV (PTVO, 29–96 cm^3^). For all patients, three plans were made in a non-clinical research version of the Eclipse treatment planning system with the aid of the Python eclipse scripting application programming interface (PyESAPI, Varian Medical Systems, Palo Alto, CA, USA): (1) photon VMAT-plans, (2) IMPT-plans and (3) proton TB-plans. The prescription for all plans was 35 fractions of 2/1.55 Gy to PTVB (70 Gy)/PTVE (54.25 Gy). Planning aimed to achieve low mean doses to both parotid glands, submandibular glands, oral cavity, individual swallowing muscles (pharyngeal constructor muscles, cricopharynx, larynx, upper esophageal sphincter [UES] and esophagus) and keep maximum spinal cord and brainstem dose <50 Gy.

Two-arc photon VMAT-plans were made using RapidPlan™ (Varian Medical Systems) [30]. The IMPT-plans used a clinically acceptable three-beam arrangement (gantry angles 45°, 180° and 315°), previously associated with satisfactory plan quality [31,32]. Since TB-plans deliver a dose along the spot track that varies relatively little with depth and do not benefit from the Bragg-peak, a larger number of beams will be needed to achieve sufficient OAR sparing. We used 10 coplanar equidistant beams (gantry angles 279° to 81°, separated by 18°; fewer beams led to a decrease in plan quality). Beam energy was 244 MeV, sufficient to, in most cases, place the Bragg-peak behind the body (in some rare cases, spots were placed in the shoulder) and irradiate the body and target with the proximal beam segment for most spots. Spot weights for both proton plans were optimized using multi-field optimization (MFO) with objectives for PTVB, PTVE, PTVO, multiple OARs, and several rings around the PTVs to avoid hotspots. For the IMPT-plans line objectives were set in the optimizer using RapidPlanPT™ (Varian Medical Systems) with a 50-patient, 3-field IMPT plan model [33]. The same RapidPlanPT™ predictions were used for TB-plans, with the exception that mean dose objectives were used for most OARs, since the TB optimizer does not accept line objectives. By decreasing RapidPlanPT™ mean objectives by 15% (except for mean objectives ≥56 Gy to avoid pulling too hard on OARs overlapping PTVB) plan quality was found to be similar to plans using line objectives. After optimization, spots with monitor units (MU) below a minimum MU threshold were removed. Minimum MU values of 1 and 10 were used for IMPT and TB-plans, respectively (see below). The quality (dosimetry) of VMAT, IMPT and TB-plans was compared using dose volume histograms (DVH) and PTV and OAR dose statistics.

### 2.2. UHDR Analyses

Two parameters were considered: (1) dose-rate and (2) beam-on irradiation time [27,28,34]. The analyses are reported for one fraction of 2 Gy and consider the depth dose profile of the beam proximal to the Bragg-peak as constant over the limited path length in the head and neck (see also Appendix A; a single spot has ~18% dose variation over 15 cm). The results are calculated based on the two-dimensional cross-section data along the beam line assuming similar results with depth and do not correspond to a specific location or tissue within the patient.

#### 2.2.1. Dose-Rate

Earlier single-fraction, single-beam UHDR research defined dose-rate as delivered dose per unit time. This is different for pencil-beam scanning (PBS) delivery, in which multiple spots are delivered sequentially. Voxels can receive dose from multiple beams, and from multiple spots/beam and therefore at multiple dose-rates, leading to voxel-specific dose-rate distributions. To determine these, the spot dose and spot dose-rate were each modelled as a circular Gaussian with sigma = 0.45 cm. The width of the pencil beam varies slightly depending on penetration depth, and sigma = 0.45 cm corresponds to the average width of a 244 MeV pencil beam at a water equivalent depth of 4–9 cm, which is a typical tumor depth. The maximum dose-rate is at the spot center: spot peak dose-rate (SPDR). Using PyESAPI, data such as (x,y) spot-center coordinates, spot irradiation times and spot MU were obtained and Matlab (MathWorks, version R2018b, Natick, MA, USA) was used to model spot dose and dose-rate. The (x,y)-specific dose-rate distribution within a beam was then determined by calculating for each contributing spot the dose share of the total dose at (x,y) and its dose-rate. At each location (x,y) within the beam, the UHDR-contribution was also calculated, defined as the percentage of local dose delivered at dose-rates ≥40 Gy/s:
(1)$$\begin{array}{c} UHDR_{c}\left(x,y\right)=\left(\underset{i\;:\;dr_{i}\left(x,y\right)\geq 40}{\sum }\frac{\;D_{i}\left(x,y\right)}{\;D_{tot}\left(x,y\right)}\right)\cdot 100\% \end{array}$$with dr_i_(x,y) the dose-rate (Gy/s) of spot i at location (x,y), D_i_(x,y) the dose (Gy) delivered by spot i at location (x,y) and D_tot_(x,y) the total dose at location (x,y), which is the sum of all spot doses at (x,y). Additionally, the dose-averaged dose-rate (DADR) was calculated for each location (x,y), giving the dose-rate averaged over all spots, weighted by the dose share of each spot to the location [2]. To only include locations receiving some dose from the beam, a total dose threshold was used, varying between 0.5–4 cGy. The dose-rate calculations did not take into account the scanning time of the beam and consider only the dose-rate contributions of all different spots contributing at least 0.5–4 cGy to a location.

#### 2.2.2. Beam-on Irradiation Time

Beam-on irradiation times are the sum of all individual spot times. To determine the location-specific irradiation time, the times of distant spots contributing negligible dose (<0.5 cGy) were excluded. The total plan beam-on time is the sum of all beam times (all spots of all beams). As for the spots, not every location is irradiated by each beam, especially outside the PTV, in which case their total irradiation time will be lower.

### 2.3. Plan Splitting

The UHDR-related parameters described above depend on SPDR, which is directly proportional to gantry current. In this version of Eclipse, the gantry current is scaled to deliver the minimum spot MU at the minimum spot time (and therefore maximum dose-rate). Since the minimum spot time is a fixed machine parameter, assumed to get as low as 0.5 ms [35], gantry current (and therefore SPDR) are maximized by increasing spot MU^2^. However, increasing the minimum spot MU degrades plan quality, due to elimination of lower MU spots during dose calculation. This is the reason for setting the minimum MU value of the TB-plans to 10: the quality remains sufficient, while increasing the maximum possible SPDR to ~48 Gy/s, exceeding our UHDR-threshold. In order to increase SPDR further, the plan can be divided into three subplans (total of 30 beams), each consisting of spots with different MUs: (1) 10–20, (2) 20–40, (3) ≥40 (Appendix A). These subplans can be successively delivered (yielding the original plan) using different gantry currents, resulting in SPDRs of ~48 Gy/s, ~93 Gy/s and ~186 Gy/s, respectively. Both UHDR-parameters were evaluated for the original (non-split) and split plans.

## 3. Results

### 3.1. Quality Comparison

All ten patients had satisfactory TB-plans with the RapidPlan-model generated objectives and 1–2 optimization rounds. Final TB-plans had OAR doses generally comparable with IMPT-plans, with <2 Gy max/mean dose difference for 12 out of 16 OARs (Table 1, Figure 1). While average oral cavity mean dose was 4.1 Gy higher for TB than IMPT, this was still considerably less than VMAT (TB: 29.3 Gy, IMPT: 25.2 Gy, VMAT: 38.6 Gy; difference with TB significant with *p* < 0.0001). TB mean body (integral) dose was also higher than IMPT, but again lower than VMAT (TB: 9.4 Gy, IMPT: 6.9 Gy, VMAT: 11.4 Gy; difference with TB significant with *p* < 0.0001).

### 3.2. Dose-Rate

Figure 2 shows the results for original TB-plans, TB split-plans and the three subplans, for a total dose threshold of 2 cGy. Figure 2A shows the average dose-rate distribution of all locations of all 100 beams (of all 10 patients) for each subplan category. For subplans with spot-MU ≥40, 20–40 and 10–20, the average percentage of dose delivered at UHDRs was 69%, 49% and 17%, respectively. Figure 2B shows a similar result for the UHDR-contribution (percentage of dose delivered at UHDRs, ≥40 Gy/s): for subplan MU ≥40, 61%/45% of the locations have a UHDR-contribution of ≥80%/≥90%. This is higher than subplans with MU 20–40 (24%/13%) and 10-20 (9%/6%). Figure 2C-E shows that the total split plan (sum of the subplans), delivers on average more dose at UHDRs (46%) and has a higher DADR (46 Gy/s) than the non-split plan (12%, 21 Gy/s). Additionally, Figure 3 shows a larger green beam area (UHDR-contribution ≥80%) and smaller red beam area (UHDR-contribution ≤10%) for the split plan than for the non-split plan. Low UHDR-contributions were mainly around the beam edges for both plans. For higher dose thresholds, a larger part of the beam-edge areas is excluded, leading to relatively more dose delivered at UHDRs (Table 2). In some areas the red and green regions are next to each other, which can be explained by the cut-off threshold value of 40 Gy/s and the fact that those locations are probably irradiated by only one spot.

### 3.3. Total Irradiation Time

For non-split TB-plans, spot times were 0.5–9.8ms and beam-on times, excluding scanning time, 0.12–0.82s (Figure 4). For split plans, spot times were 0.5–2.5ms and beam-on times 0.06–0.41s. Appendix A shows beam times including scanning time, as well as total irradiation times (sum of all beam times). On average, split plans have significantly lower beam-on times than non-split plans (51% lower: 1.87 s vs. 3.80 s; *p* < 0.0001), but irradiation times including scanning time increase significantly by 16% (8.86 s vs. 7.61 s; *p* < 0.0001). Beam-on times include all beam spots, but irradiation time varies within the beam, depending on the number of spots that contribute a minimum dose. Appendix A shows this variation and demonstrates how location-specific irradiation times are much lower than the beam time, with a maximum irradiation time of ~14%/~9% of the beam-on time for non-split/split plans, in a small area of the beam.

## 4. Discussion

This work has demonstrated (1) TB-plans, using 10 coplanar spot scanning proton beams, achieved comparable OAR sparing to IMPT and better than photon VMAT for fractionated HNC treatment, (2) UHDR-metrics for low fraction-doses increase with application of MU-based beam-splitting (3) irradiation times of 10-field TB-plans are short, with <4s beam-on time and <10 s total beam time including scanning.

Because TBs have their highest dose-rate at high energies, they lend themselves well to UHDR delivery. While TBs do not benefit from higher dose-rate delivery in the Bragg-peak and low dose deposition behind it (leading to a higher integral dose outside the OAR than IMPT), there are still several arguments for TBs: (1) they do not require the energy modulation needed to place Bragg-peaks in the PTV; (2) they are insensitive to density changes and change in the body contour and dimensions induced by oedema/weight loss (illustrated by patient with air in PTVB: IMPT V95% = 97.7%, V107% = 5.4% vs. TB V95% = 99.1%, V107% = 0.1%); (3) their faster delivery (no energy-modulation and increased dose-rates) may also have practical benefits: less influence of moving structures (e.g., larynx) and the facilitation of higher patient through-put—it might be possible to treat patients in multiple treatment rooms simultaneously (since the delivery time of a single beam is only ~1 s compared to >1 min for IMPT, thereby potentially also enhancing patient comfort); (4) Bragg-peak related complications, such as increases in linear energy transfer (LET) behind the Bragg-peak (e.g., linked to pediatric brainstem necrosis [36,37]) are not relevant for TBs, and (5) dose verification may be possible using distally placed detectors.

### 4.1. Plan Quality

In earlier work [1,3] acceptable TB-plans were achieved for lung SBRT and we have now also demonstrated this for HNC, with larger tumors and more OARs (Table 1). TBs achieved for all OARs and for the entire body considerably lower mean doses than VMAT, and for most OARs, comparable mean doses as IMPT. Only the mean doses to the oral cavity and the integral body were >2 Gy higher for TB-plans than for IMPT.

The higher integral dose in TB-plans is mainly due to the fact that beams do not stop in the target, and also deliver dose behind it. A small amount can also be explained by Bragg-peaks in the shoulders. However, usually only a few spots are placed in the body (outer beams only) and they are often not close together minimizing local dose accumulation (Appendix A). An explanation for favorable TB-plan quality is the steeper TB penumbra (Appendix A). One limitation is that the comparative IMPT-plans were not made using robust optimization, however, we have also not quantified the robustness advantage of TB-plans for density changes in internal anatomy.

### 4.2. Dose-Rate

For the purpose of these analyses, we looked at dose-rates above 40 Gy/s. However, the true threshold for the FLASH-effect may differ from this [27,28] and it may even be the case that there is no binary threshold, and that FLASH-effect increases with dose-rate [9] and fraction size [38]. While it seems probable that the FLASH-effect increases for higher dose/fraction and higher dose-rates it is not known if no effect exists at lower fraction sizes and dose-rates. Even a small effect for lower doses/fraction could be clinically useful and add to the other advantages of conventionally fractionated TB radiotherapy. If evidence for a FLASH-effect at low fraction sizes accumulates, then we have shown that TBs are capable of achieving a reasonable UHDR-contribution.

It should be noted that dose-rates can be calculated in different ways [2,3,8,27,34], of which we have chosen two: (1) a dose-rate distribution based on the SPDR of the contributing spots in one beam and not taking scanning time into account, (2) DADR, which is also based on SPDR and ignores dead time between spots. We did not consider the dose-rate averaged over the time needed to deliver all the contributing spots, nor the dose-rate averaged over all the spots from all beams, contributing to each voxel. This has likely resulted in an overestimation of the dose-rate. However, since it is currently not yet known which definition of dose-rate is important for the FLASH-effect, it is difficult to decide which method to favor. The work described here can easily be adapted as knowledge accumulates and the overall conclusion holds: that at each voxel dose is delivered at multiple dose-rates, the biological effect of which is currently unknown.

The UHDR-contribution is influenced by SPDR which is limited by a minimum spot time of 0.5 ms, currently feasible on certain machines [39], but not yet clinically available. Higher minimum spot times will decrease SPDR resulting in little to no dose delivery at UHDRs. SPDR is also dependent on minimum spot MU, which is currently not considered by the optimizer, leading to lower quality TB-plans than possible. Future incorporation of spot MU into optimization could increase the proportion of UHDR delivery. Another way of achieving this would be for the optimizer to maximize high MU spots, for example by spot reduction techniques [2], or by incorporating dose-rate in the optimization [40]. We split plans depending on spot MUs to increase UHDR-metrics: by delivering higher MU spots with a higher gantry current, more dose can be delivered at UHDRs (Figure 2).

### 4.3. Total Irradiation Time

As for dose-rate, irradiation time definition is unclear when considering spot-by-spot scanning. The influence of “dead time” between pulses on the FLASH-effect is currently uncertain, so for the purposes of this analysis, we considered both beam-on time and irradiation time including scanning (for a single beam and not the entire plan). FLASH has been associated with published single-field irradiation times in the order of <0.2 s [27,28,34] to ~0.5 s [7]. Our maximum beam-on time (one beam direction) for split plans was 0.41s and even lower when considering local irradiation time. Although spot and beam-on times are lower for split plans, irradiation time including scanning increases by ~16%, because of the triple scanning of each beam. However, even with the addition of switching time between gantry currents, assumed to take <1 s/switch, total delivery time is still much lower (original plan < 20 s, split plan < 40 s) than the several minutes typical of a three-field HNC IMPT-plan [41] (mostly due to energy layer switching, typically several seconds/switch [42]). Note that calculated beam-on times are the sum of the spot times, even though the entire area is not irradiated by each spot. Location-specific beam times depend on the number of spots irradiating a given location and are therefore much lower (Appendix A).

Both the dose-rate and irradiation time analyses are influenced by the fact that we are dealing with multiple spots. This issue can be avoided by passively scattered beams, that have been shown to demonstrate the FLASH-effect [14]. However, the dose-rate of such beams drops rapidly with field sizes, making them unsuitable for the relatively large tumors described here.

The proposed transmission beam plans are all coplanar and we do not expect much difference in delivery between photons and protons. Photon plans consist of two full arcs, each delivering their dose continuously in ~70 s. Proton plans consist of 10 static fields, equidistantly placed over a half arc where each field is delivered in <1.8 s and the gantry rotation speed is similar to photon therapy. For both treatments, patient setup can be done using a CBCT scan. Currently, there are no photon FLASH methods available that can deliver such treatments, although photon FLASH machines are under development [43].

## 5. Conclusions

In this technically focused paper, we have demonstrated that: (1) complex HNC TB-plans achieve dosimetry to the PTV and most OARs that are comparable to IMPT-plans and considerably better than VMAT. They are simpler, faster to deliver and in some situations more robust than IMPT plans, making them a clinically relevant delivery modality and deserving of more attention; (2) more radiobiological research is needed to investigate the exact conditions of FLASH and the mechanism behind it [44], but if some FLASH-effect is possible at conventional, relatively low, fraction doses, then we have shown that TB-plans are well positioned to capitalize on this.

## Figures and Tables

**Figure 1 cancers-13-01859-f001:**
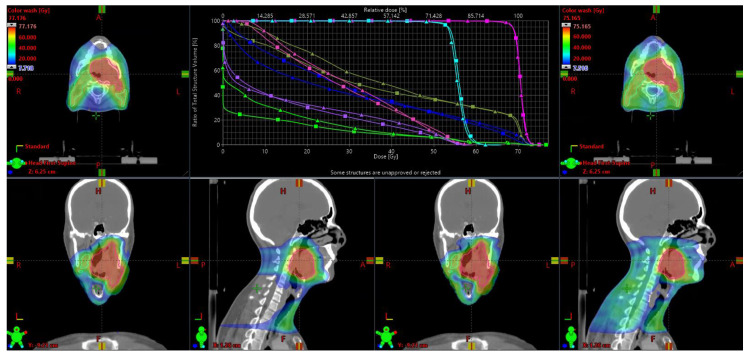
Comparison of dose distribution and dose volume histograms (DVH) for IMPT (left, squares) and TB (right, triangles) plan. Bright pink = boost planning target volume (PTVB), light blue = elective planning target volume (PTVE), green = body, purple = contralateral parotid, blue = upper larynx, pink = contralateral submandibular gland, moss green = oral cavity.

**Figure 2 cancers-13-01859-f002:**
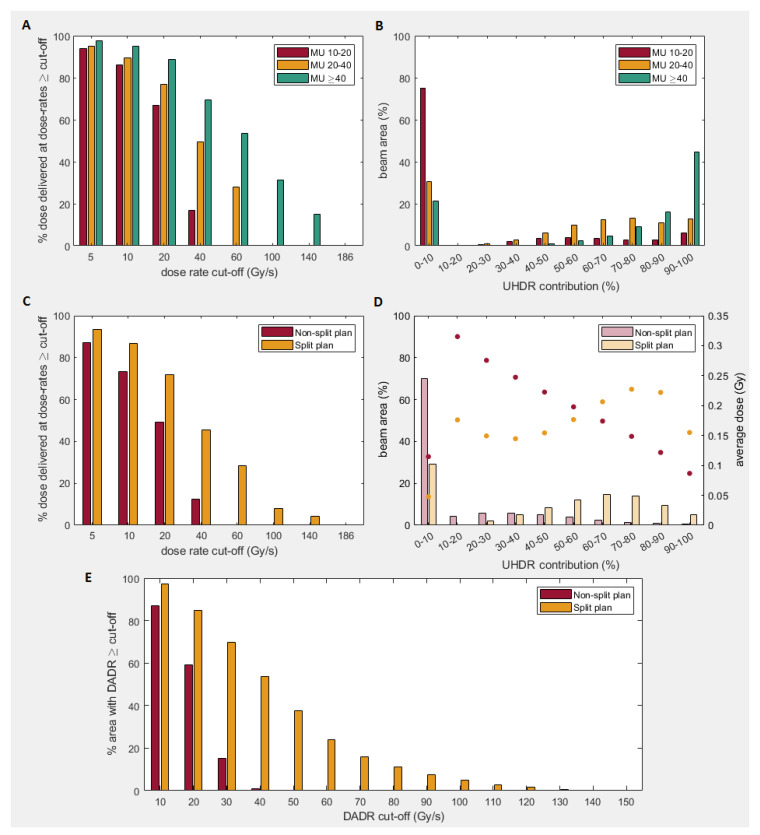
Average dose-rate distribution (**A,C**) and percentage ultra-high dose-rate (UHDR) contribution (**B,D**) for the subplans (upper) and the non-split/split plan (lower). In B, D, the UHDR-contribution (≥40 Gy/s) is shown vs. percentage beam area. D also gives the average dose for each UHDR-contribution interval. (**E**) gives dose-averaged dose-rate (DADR) distribution for non-split/split plans. The average DADR was 46/23 Gy/s for the split/non-split plans. Total dose threshold is 2 cGy.

**Figure 3 cancers-13-01859-f003:**
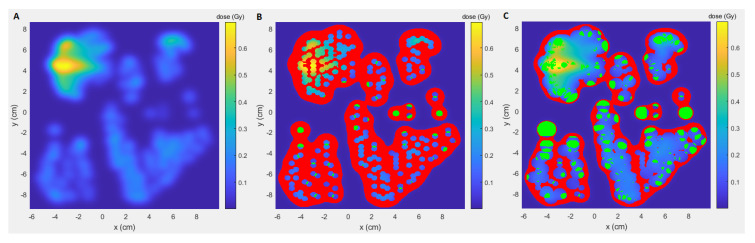
Dose within cross-section of a beam (**A**), with points with UHDR-contributions ≤10% (red) and ≥80% (green) for non-split (**B**) and split plan (**C**). Total dose threshold is 2 cGy.

**Figure 4 cancers-13-01859-f004:**
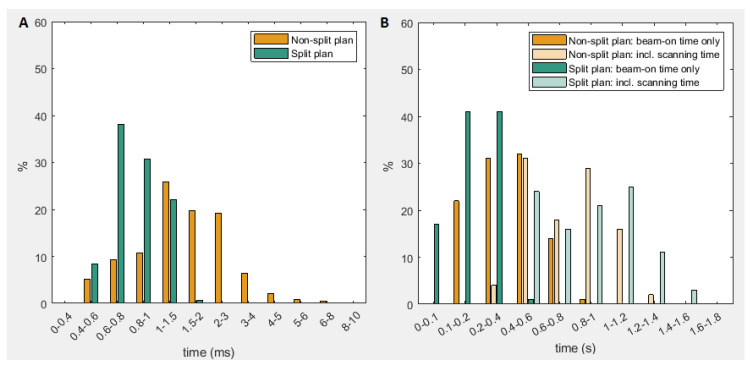
(**A**) spot time distribution for all spots of all 100 beams; (**B**) beam time distribution for all 100 beams.

**Table 1 cancers-13-01859-t001:** Dose statistics volumetric-modulated arc therapy (VMAT), intensity-modulated proton therapy (IMPT) and transmission beams (TB) plans. Average target dose percentages for target structures (Pavg) and average doses (Gy) for organs-at-risk (OARs) (Davg), difference between TB and IMPT/VMAT (∆Pavg / ∆Davg). SMG = submandibular gland, UES = upper esophageal sphincter, c = contralateral, i = ipsilateral.

Structure	Metric	P_avg_ VMAT	P_avg_ IMPT	P_avg_ TB	∆P_avg_ (TB-IMPT)	∆P_avg_ (TB-VMAT)
PTVB	D_mean_ V_95%_	102.4 99.3	101.2 98.7	100.8 99.2	−0.3 ^†^ 0.5	−1.5 −0.1
	V_107%_	2.2	0.7	0.0	−0.7	−2.2
PTVE	D_mean_	104.4	103.9	102.7	−1.2 ^†^	−1.7 ^†^
	V_95%_	98.8	97.4	98.1	0.6 ^†^	−0.9 ^†^
		D_avg_ VMAT	D_avg_ IMPT	D_avg_ TB	∆D_avg_ (TB-IMPT)	∆D_avg_ (TB-VMAT)
Spinal cord	Dmax	42.7	36	40.6	4.6	−2.0 †
Brainstem *	Dmax	38.4	23.8	35.3	11.5	−3.1
Oral cavity	Dmean	38.6	25.2	29.3	4.1 †	−9.3 †
Cricopharynx	Dmean	23.2	13.7	12.5	−1.3 †	−10.8 †
Esophagus	Dmean	16.7	11.5	10.2	−1.3 †	−6.5 †
SMG (c) **	Dmean	35.1	25.2	26.6	1.4 †	−8.5 †
SMG (i)	Dmean	62.3	59.7	60.2	0.5	−2.1
Lower larynx	Dmean	20.2	10.2	9.4	−0.8 †	−10.8 †
Upper larynx	Dmean	37.8	29	27.8	−1.2	−10.1 †
Parotid (c)	Dmean	21.0	12.8	13.4	0.5	−7.6 †
Parotid (i)	Dmean	25.3	18.4	18.4	0.1	−6.9 †
Pharynx Inf	Dmean	27.6	16.6	15.4	−1.1 †	−12.1 †
Pharynx Med	Dmean	50.4	39.8	39.3	−0.5	−11.1 †
Pharynx Sup	Dmean	57.4	52.3	52.4	0.1	−5.0 †
UES	Dmean	15.5	9.3	7.4	−1.9 †	−8.1 †
Body	Dmean	11.4	6.9	9.4	2.5 †	−2.0 †

* data for 7/10 patients with contoured brainstem. ** data for 9/10 patients with contoured contralateral submandibular gland. † *p*-value < 0.01, paired t-test with alpha = 0.05.

**Table 2 cancers-13-01859-t002:** Dose-rate results for multiple total dose thresholds. DADR = dose-averaged dose-rate, NS = non-split plan, S = split plan, MU = monitor unit. Data for the split plan will be influenced by the relative proportion dose delivered at different MU thresholds. All differences between non-split and split plans are significant, with *p* < 0.0001 (paired t-test, alpha = 0.05). All data are pooled and results are averaged for all beams from all plans.

Threshold (cGy)	DADR (Gy/s) Mean(std)		% Dose Delivered at UHDR Mean(std)					UHDR-Contribution ≥80%/≥90% (%) Mean(std)				
	NS	S	MU 10–20	MU 20–40	MU ≥40	NS	S	MU 10–20	MU 20–40	MU ≥40	NS	S
0.5	18.3 (0.6)	39.4 (4.2)	10 (0.4)	38 (2.1)	50 (2.1)	10 (0.8)	38 (4.2)	5 (1.0)/ 3(1.1)	18 (2.6)/ 10 (2.8)	44 (1.6)/ 32 (4.9)	1 (0.6)/ 1 (0.4)	12 (2.6)/ 4 (1.6)
1	19.6 (0.4)	42.3 (3.9)	12 (0.4)	42 (1.4)	58 (1.3)	11 (0.6)	41 (3.7)	6 (1.5)/ 4(1.5)	20 (3.4)/ 11 (3.4)	51 (2.9)/ 37 (6.7)	1 (0.6)/ 1 (0.4)	13 (2.9)/ 5 (1.9)
2	21.3 (0.4)	46.2 (3.6)	17 (1.2)	49 (1.1)	69 (2.5)	12 (0.4)	46 (3.1)	9 (2.6)/ 6(2.5)	24 (4.9)/ 13 (4.5)	61 (5.6)/ 45 (9.8)	1 (0.8)/ 1 (0.5)	15 (3.5)/ 5 (2.2)
3	22.4 (0.5)	49.1 (3.5)	23 (2.8)	55 (2.3)	79 (4.5)	14 (0.5)	50 (2.7)	11 (4.1)/ 7 (3.7)	27 (6.2)/ 14 (5.4)	69 (8.1)/ 51 (12.5)	2 (0.8)/ 1 (0.5)	16 (4.0)/ 6 (2.5)
4	23.2 (0.6)	51.4 (3.6)	28 (4.6)	60 (3.0)	83 (4.7)	15 (0.6)	53 (2.7)	11 (5.2)/ 7 (4.5)	28 (6.6)/ 14 (5.5)	72 (8.6)/ 52 (13.2)	2 (0.9)/ 1 (0.5)	17 (4.3)/ 6 (2.5)

## Data Availability

Restrictions apply to the availability of these data. Data was obtained from Varian Medical Systems and are available from the authors with the permission of Varian Medical Systems.

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
