# Peer review of "Ultra-High Dose Rate Transmission Beam Proton Therapy for Conventionally Fractionated Head and Neck Cancer: Treatment Planning and Dose Rate Distributions"

_cancers, 2021, doi:10.3390/cancers13081859_

Round 1

Reviewer 1 Report

Good presentation! Compliment

Author Response

Thank you for your positive feedback.

Reviewer 2 Report

This is an interesting study about ultra-high dose rate transmission beam proton therapy for conventionally fractionated head and neck cancer. The authors observed that TB-plan quality was comparable to IMPT, for contoured organs at risk, and better than VMAT. Any potential FLASH-effect would only further improve plan quality. TB-plans can also be delivered quickly, which might facilitate higher patient through-put and enhance patient comfort.

The paper is well written. However, some issues remain.

Please add some data about TNM staging of the 10 patients and better specify the site of tumors (tonsil, tongue base, pyriform sinus, etc.).

The authors stated that oral cavity mean dose was higher for TB than IMPT, but considerably less than VMAT, and that TB mean body (integral) dose was also higher than IMPT, but again lower than VMAT. However, no statistical analysis was performed. Please add it and report p values.

Statistical analyses should be performed also for other parameters reported in the study.

Author Response

Thank you for your feedback. We have made certain changes to the manuscript, which are indicated in yellow.

TNM staging and location have been added to the first paragraph of the Materials and Methods.

We have performed additional statistical analyses. In Table 1 the significant differences between TB and IMPT/VMAT were denoted by the dagger symbol and we have included the most important p-values in the text. In Table 2 the standard deviation has been added for all values and we performed t-tests to analyze the differences between split and non-split plans.

Reviewer 3 Report

The Authors report on a feasibility study evaluating, with a planning comparison design, the potential use of  ultra-high dose rate transmission beam proton therapy to deliver conventionally fractionated radiotherapy in the clinical setting of head and neck cancer. The researchers compared transmission beam proton therapy to a) intensity-modulated proton therapy (IMPT) and b) photon-based volumetric modulated arc therapy (VMAT) in terms of PTV coverage and OARs sparing. They also evaluated the possibility to obtain the so called ‘FLAS-effect’ at a daily dose of 2 Gy, as employed to conventional fractionation. I think the topic is of interest. The study is well-designed, the methodology is scientifically sound and the manuscript is well-written. My only suggestion would be to add, in the discussion section, few lines to comment on the eventual differences (in terms of advantages and disadvantages) in the delivery of FLASH with protons compared to photons in the setting of HN cancers.

Author Response

Thank you for your feedback. We have made certain changes to the manuscript, which are indicated in yellow.

A few sentences on the differences in FLASH delivery between protons and photons have been added to the end of the discussion.

Round 2

Reviewer 2 Report

Thank you for improving the manuscript.